# The Influence of Yeast Strain, β-Cyclodextrin, and Storage Time on Concentrations of Phytochemical Components, Sensory Attributes, and Antioxidative Activity of Novel Red Apple Ciders

**DOI:** 10.3390/molecules24132477

**Published:** 2019-07-05

**Authors:** Sabina Lachowicz, Jan Oszmiański, Martyna Uździcka, Joanna Chmielewska

**Affiliations:** 1Department of Fermentation and Cereals Technology, Wrocław University of Environmental and Life Science, 37 Chełmońskiego Street, 51-630 Wroclaw, Poland; 2Department of Fruit, Vegetable and Plant Nutraceutical Technology, Wrocław University of Environmental and Life Science, 37 Chełmońskiego Street, 51-630 Wroclaw, Poland; 3Department of Food Technology and Human Nutrition, Rzeszów University, 4 Zelwerowicza Street, 35-601 Rzeszów, Poland

**Keywords:** polyphenolic compounds, antioxidative properties, organic acids, *Saccharomyces cerevisiae*, *Saccharomyces bayanus*, principal component analysis (PCA)

## Abstract

The yeast strain and storage time is an important factor affecting the development of phytochemicals and sensory attributes in ciders. Therefore, the aim of this study was to determine the influence of yeast strains (*Saccharomyces bayanus* and *Saccharomyces cerevisiae*), *β*-cyclodextrin (BCD), and storage time on physicochemical parameters, contents of phenolic compounds (ultra-performance liquid chromatography with photodiode array detector coupled to quadrupole time-of-flight tandem mass spectrometry (UPLC–PDA–QToF-MS/MS)), antioxidative activity (free radical-scavenging ability (ABTS) and ferric reducing antioxidative power (FRAP) assay), and sensory attributes of new cider from the “Bella Marii” cultivar of red apple. The pH value, acidity, concentrations of alcohol, organic acids, and polyphenols; and the color and antioxidative properties were evaluated in red apple ciders immediately after fermentation and after three months of storage at 4 °C. *S. cerevisiae* SIHAFERM Finesse Red with BCD and SIHAFERM Finesse Red yeast strain especially contributed to obtaining ciders with a high content of the tested compounds. The use of BCD during fermentation significantly influenced the protection of bioactive compounds, by as much as 18%. Storage time had an impact on concentrations of the tested components (mainly on the total flavan-3-ols and phenolic acids). Based on the achieved values of parameters analyzed in red apple ciders and results of the consumer acceptance test, it may be concluded that red apple offers vast potential for the production of ciders with a high content of polyphenolic compounds.

## 1. Introduction

Cider is a fermented beverage with alcohol content from 1.2 to 8.5% *v*/*v* made mainly of apples. It should be emphasized that its quality and character are determined foremost by the apples used in its production process. Special attention should be paid to the contents of polyphenols, acids, tannins, and sugars in the apples, because they are responsible for the sensory characteristics of this fermented beverage [1]. The polyphenolic compounds, including anthocyanins, contribute to the development of mainly color, bitterness, and astringency. They may also influence the sweetness and sourness of food products, especially of alcohol products [2,3]. Moreover, they can influence the aroma of ciders via enzymatic decarboxylation during fermentation, due to the volatile phenolics formed. Furthermore, polyphenols exhibit multiple health-promoting benefits, including strong anti-inflammatory, antioxidative, anti-viral, and anticarcinogenic properties. They are implicated to reduce the risk of diabetes, cardiovascular diseases, Alzheimer’s disease, atopic dermatitis, and cancer. Another important component of ciders is ethanol. It is a key contributor to the quality, and fundamental to the sensory properties and stability of low-alcohol beverages. In addition, it acts as a solvent for aromas and dyes. Cider production is feasible through both the use of pure strains of microorganisms, as well as spontaneous fermentation [1], which offers a large variety of microorganisms. Spontaneous fermentation imparts an interesting taste, complexity, and unusual features to the product; nevertheless, it can be a difficult process due to its limited control and repeatability [2]

The selection of yeast cultures is important for the proper course of a fermentation aimed at manufacturing products with desirable sensory attributes, health-promoting properties, and high quality [1]. Research results suggest the high potential of using associated yeast cultures, not only from the *Saccharomyces* genus, for the production of fermented beverages [3]. However, other strains of yeast are not as resistant to increasing ethanol concentration, which makes the fermentation process difficult. Therefore, the selected strains of *Saccharomyces cerevisiae*, *Sachchramyces bayanus*, or *Saccharomyces paradoxus* yeast are commonly used in this process as they adapt very well to growth in environments with low pH (2.9–3.8) and a high sugar content (200–300 g/L). The selection of the appropriate strain, and checking its influence on the apple cider quality and also on its profile of polyphenolics are extremely important as it affects the sensory characteristics of the final product [4,5,6,7].

In addition, BCD inclusion complexes are used on a large scale in the food industry to protect and stabilize substances sensitive to moisture, light, storage time or oxygen like for example anthocyanins. BCD is a cyclic oligosaccharide consisting of seven glucose molecules joined by an α-1,4-glycosidic bond. BCDs are also used to mask the undesirable color, odor, and taste of selected food products or to bind volatile and highly toxic substances, often leading to a significant extension of storage time [8,9]. Furthermore, BCD inclusion complexes with hydrophobic molecules can milden the bitter taste by inhibiting the interaction between the bitter receptor and the molecule [10]. 

Therefore, the aim of this study was to evaluate the influence of selected yeast strains (*S. bayanus* and *S. cerevisiae*), BCD (with *S. cerevisiae*, with SIHAFERM Finesse Red), and storage time (three months at 4 °C) on physicochemical parameters, concentrations of phenolic compounds, and the antioxidative activity of a new cider from the “Bella Marii” cultivar of red apple. An additional objective was to determine the impact of the factors used on the sensory attributes of the red apple ciders. We assumed that the combination of SIHAFERM Finesse Red yeast - responsible for the protection of color, and BCD - responsible for the protection of polyphenolic compounds, especially anthocyanins, would allow producing ciders with the highest biological potential.

## 2. Results and Discussion

### 2.1. Physicochemical Parameters and Fermentation Kinetics in Red Apple Cider

Results of the determinations of the basic chemical parameters of the investigated red apple cider before and after fermentation are presented in Table 1 The fermentation of red apple cider prepared from the “Bella Marii” cultivar was performed at 20 °C for three weeks. Alcohol content in the red apple cider after fermentation and after three months of storage at 4 °C was not significantly affected by storage time or by the yeast strain added. A similar observation was made by Versari et al. [11] for the prepared grape wine, the parameters of which remained unaffected by the yeast strain used for its fermentation. Ciders are divided with respect to ethanol content, which ranges from 1.2% in sweet ciders to 8.5% in dry ciders. The ethanol value in red apple cider was not equal and ranged from 6.88 vol.% in the cider with *Saccharomyces cerevisiae* (SIHAFERM Finesse Red with the addition BCD—dry cider) to 6.92 vol.% in the cider with *Saccharomyces cerevisiae* (Pure Nature—dry cider). Štornik et al. [12] determined similar contents of alcohol in organic and conventional apple cider vinegar (from green apple) of around 6.0 and 6.3 vol.%, respectively. 

The pH value and the total acidity of red apple ciders before and after storage at 4 °C were not significantly affected by the yeast strain added. Their mean pH value was 3.28 and their total acidity was 6.37 g/L. The storage time had a significant effect on the total acidity, which ranged from 6.25 g/L before storage to 6.51 g/L after storage (*p* < 0.05), and these results were similar to the acidity determined in ciders from green apple accounting for 6.70 g/L on average [13]. The mean content of total organic acids determined in red apple cider was not significantly (*p* < 0.05) affected by the yeast strain used, and it reached 7865.52 mg/L (Table 2). In contrast, storage time had a significant effect on the content of both total organic acids and individual acid (*p* < 0.05). Among the eight organic acids analyzed in red apple cider, the main ones were malic acid (69.5%)  >  citric acid (16.6%) >  oxalic acid (4.2%) > quinic acid (4.1%) > succinic acid (<2.8%)  ≥ maleic, fumaric, and acetic acid (<0.2%). The major acid in red apple cider before and after storage turned out to be malic acid, which represented around 90% of the total acidity [13,14]. Its content remained stable regardless of yeast addition and storage time and was on average 4.4 times higher than in green apple fruit [15]. In turn, the addition of BCD with *S. cerevisiae* SIHAFERM Finesse Red to the red apple cider caused an increase by 1.3 and 1.8% and a decrease by 1.3% in the contents of oxalic, fumaric, and citric acids, respectively. The organic acids identified in ciders may originate directly from the raw material or may be formed during fermentation. The first group of acids includes malic, citric, oxalic, quinic, maleic, and fumaric acids, contents of which depend on the cultivar, climate, region, and degree of apple ripeness. The acids appearing after fermentation include succinic, acetic, and lactic acid. The total content of organic acids in cider determines their acidity, which is an important trait responsible for cider color and for the course of the fermentation process. Furthermore, the total acidity and the content of acids have a direct impact on the flavor of ciders [13,16,17]. In turn, the low pH values and high contents of organic acids impact their microbiological stability [17]. This, it is important to test and check the impact of yeast strain and storage time on the values of these parameters in ciders. Moreover, these parameters improve the proper function of the body through regulate the proper chemical reactions and the secretion of digestive enzymes [17].

Color was one of the main quality attributes tested in red apple ciders after fermentation and after storage at 4 °C (Table 1). Generally, the appearance of beverages and food products affects their choice by consumers, whose decisions are driven by such criteria as overall acceptability, sensory characteristics, and safety. The main problem of apple ciders is their browning [17,18]. The red apple ciders were evaluated, before and after storage, for color parameters lightness (L*), yellowness (b*), and redness (a*), and also for chroma difference (ΔC), and total color difference (ΔE) (Table 1). Values of color components determined for the red apple juice were L* = 65, a* = 45, and b* = 53. In the case of ciders, color parameters differed significantly depending on the type of yeast used, storage time, and BCD addition (*p* < 0.05). Statistically significant differences were determined for values of the L* parameter, corresponding to brightness, in all ciders. The use of yeast *S. bayanus* during fermentation contributed to on average 1.3 times higher brightness of the samples compared to the use of *S. cerevisiae*. In turn, the red apple cider was 1.2 times brighter than the samples before storage. The lowest value of the L* parameter was recorded in red apple cider fermented with *S. cerevisiae* (SIHAFERM Finesse Red with BCD and Rubino Cru). The use of BCD resulted in the darkening of the samples by 1.6 and 1.9 times on average before and after storage, respectively. The value of color parameter a* increased, whereas that of parameter b* decreased in all samples of red apple cider after fermentation and maturation. It was indicated by a less yellow and more red color of the final product. The yeast strains, storage time, and BCD addition influenced the L*, a*, and b* values to various extents. The red apple cider fermented with *S. cerevisiae* (SIHAFERM Finesse Red with BCD and SIHAFERM Finesse Red) was characterized by the highest value of parameter a*, which was indicated by its more intense red color. The use of these yeasts with or without BCD improved cider color during fermentation as well as after treatment. These ciders were characterized by an intense red and attractive color compared to these made with the other yeast tested. The ΔE parameter allows distinguishing colors of two products when ΔE ≥ 5 units [19]. It is a valid parameter for the industry, and, in the analyzed red apple ciders, its values depended on storage time and yeast strain. Its highest value (ΔE > 5 units) was determined for the cider fermented with *S. cerevisiae* (especially SIHAFERM Finesse Red with BCD). The ΔC parameter defines the chromaticity, being a measure of color intensity or saturation, and its values range from 0 (completely unsaturated) to 100 or more (pure color) [20]. In the present study, ΔC significantly depended on yeast strain and storage time, and its highest value was found for the ciders fermented with *S. bayanus* (25.49), and the lowest one for the sample with the addition of BCD (9.73). In red apple ciders, the ΔC average value was 1.4 times higher than in the ciders before storage.

The results of the fermentation kinetics analysis are illustrated in Figure 1. It was found that the fermentation of SIHAFERM Finesse Red with the addition of BCD was the fastest, with a weight loss of 24.54 g after 24 h. However, the remaining yeast cultures started fermentation later and its kinetics was slower values. The kinetics of fermentation depends on such factors as acidity, temperature, sugar content in the raw material, and the concentration of ethyl alcohol increasing during the process [21,22]. The present research shows fermentation kinetics to range from 5.21 to 5.44 g/100 cm^3^. In contrast, in the study of [19], the kinetics of apple musts Satora fermentation with monocultures and mixed cultures was from 9.14 to 10.07 g/100 cm^3^ [23]. This difference results from the use of other apple cultivars characterized by a different acidity and total sugar content. However, studies by Boudreau et al. [24] showed that commonly used fungicides remained on apples and contributed to the reduction of fermentation kinetics during cider production. What is more, the kinetics of fermentation depends on the yeast strain used [1]. According to literature data, yeast strains of *S. bayanus* are characterized by lower fermentation kinetics than strains of yeast *S. cerevisiae* [23]. This may be due to the low viability or activity of the yeast strain used, too high a concentration of sulfur dioxide, or the fact that *S. bayanus* strains carry out the fermentation process more effectively at low temperatures [25]. In addition, the faster fermentation of the cider with SIHAFERM Finesse Red and BCD may be due to the presence of BCD. According to Białecka-Florjanczyk and Majewska [26], BCD may interact with the substrate and reduce its concentration, affecting the course of the fermentation process, and also the type of changes occurring during fermentation [10].

### 2.2. Determination of Polyphenolic Compounds in Red Apple Cider

Phenolic compounds are very important group of bioactive compounds, because they have a strong influence on quality attributes of food products such as taste, aroma, and appearance as well as exhibit health-promoting properties (anti-diabetic, anti-inflammatory, antitumor, anti-allergic, antioxidative) [27,28,29,30,31,32]. Moreover, the cider-making process, yeast strain, and storage time are known to decrease polyphenols concentration in ciders. Furthermore, these factors may affect cider quality. Therefore, it is important to test the effect of the strains of yeast or storage time on the content of polyphenolic bonds and their protection. The identification of polyphenolic compounds extracted from red apple ciders was based on results of the ultra-performance liquid chromatography with photodiode array detector coupled to quadrupole time-of-flight tandem mass spectrometry (UPLC–QToF–MS/MS) analysis, retention time (Rt), ultraviolet–visible light (UV–vis) spectra, and data comparisons [27,28,29,30,31]. In the red apple ciders studied, 33 polyphenolic compounds were separated, belonging to the five groups that are presented in Table 3 and Table 4. The major group of polyphenols in red apple ciders were phenolic acids, including mainly hydroxycinnamic acid derivatives which accounted for 39% of all compounds. The next four different fractions of polyphenols included flavan-3-ols (36%; in this group, polymeric proanthocyanidin accounted for 24%) > anthocyanins (13%) > dihydrochalcones (9%) > flavonols (<3%). These compounds are important in ciders and/or wines because they affect their color, taste, aroma, and stability, in addition to their beneficial human effects [32]. Yeast strain and storage time had a significant (*p* < 0.05) effect on the concentration of polyphenols. The red apple ciders fermented with *S. cerevisiae* with the addition BCD were rich in the analyzed compounds whose concentrations were 1.3 and 1.2 times higher on average compared to the ciders prepared with *S. cerevisiae* and *S. bayanus*. BCD affected the stability and increased the protection of phenolic compounds, which was confirmed in stored chokeberry juice by Lachowicz et al. [9]. The lowest concentration of polyphenolic compounds was determined in the red apple cider prepared with the addition of *S. bayanus* (412.22 mg/L). The total content of phenolics decreased by around 8% after three months of storage at 4 °C, and, in the final cider, it ranged between 381.83 and 562.57 mg/L. The content of phenolic compounds analyzed by Alberti et al. [13] and Verdu et al. [28] in cider prepared from green apple was 1.5 and 2.0 times lower than in the ciders prepared from red apple. The red apple cider having the highest concentration of these compounds was prepared with the addition of *S. cerevisiae* SIHAFERM Finesse Red with BCD. In general, stabilization and protection of polyphenolic compounds, especially anthocyanins, in colored juices, like for example chokeberry juice, is achieved by the clarification with the addition of BCD because they prevent the degradation of these compounds. The protection of polyphenolic compounds is very important in the final product due to their health benefits action, therefore BCD with SIHAFERM Finesse Red yeast was used to assure the greater protection of polyphenols [9,33].

#### 2.2.1. Phenolic Acids

The first and major group of phenolic compounds in red apple ciders after fermentation by *S. cerevisiae* and *S. bayanus* and storage was phenolic acids. A total of eleven phenolic acids belonging to hydroxycinnamic acids were determined, and they were derivatives of caffeic and *p*-coumaric acids. The major components of this fraction were represented by three caffeoylquinic acid derivatives identified as 3-*O*-caffeoylquinic (Rt = 3.83 min), 5-*O*-caffeoylquinic (Rt = 4.16 min), and 4-*O*-caffeoylquinic acids (Rt = 4.30 min) with major ion [M − H]^−^ at *m*/*z* = 353, and fragmentation ion at *m*/*z* = 191. Also five isomers of *p*-coumaroylquinic acid were detected as 3-*O-p*-coumaroylquinic (Rt = 3.28 min), 4-*O-p*-coumaroylquinic (Rt = 4.81 min), and 5-*O-p*-coumaroylquinic acids (Rt = 5.12 min) with major molecular ion at *m*/*z* = 337, as well as two ethyl 3-*O-p*-coumaroylquinic acids (Rt = 6.09 and 6.33 min) showing [M − H]^−^ at *m*/*z* = 365. Two caffeic acids were also identified ([M − H]^−^ at *m*/*z* = 179). These compounds were identified using commercial standards, MS spectra, and also literature data [27,28,29,31] in green apple fruits and ciders. Significant differences (*p* > 0.05) were found in the contents of phenolic acids, affected by yeast strain and storage time. After storage, an increase by around 3% was determined in phenolic acid content in the red apple cider fermented with *S. cerevisiae* with the addition of BCD, while a decrease by 6% on average was noted in the other products. After aging, phenolic acid content in the red apple ciders ranged between 137.66 and 180.06 mg/L. According to Alberti et al. [13], hydroxycinnamic acid content determined in juice and cider obtained from green apples was 1.2 and 1.3 times lower compared to the ciders obtained from red apples. The red apple ciders fermented with *S. cerevisiae* SIHAFERM Finesse Red with the addition of BCD were characterized by the highest content of phenolic acids. In addition, the main compound in the analyzed apple ciders was chlorogenic acid, which accounted for 76% of total phenolic acids. This acid is mainly responsible for the taste values of alcoholic beverages, as well as raw material, and they exhibits antioxidative, antimutagenic, and anticancer properties. Generally, phenolic acids play an important role in the development of astringency and bitterness in wines and/or ciders [9,34]; they are also responsible for the browning of wines or beverages under the influence of oxygen. The use of *S. cerevisiae* SIHAFERM Finesse Red yeasts with or without BCD influenced the protection of phenolic acids. 

#### 2.2.2. Flavan-3-ols

The second fraction of red apple ciders was flavan-3-ols (monomers, oligomers, and polymeric procyanidins) detected as six components as (−)-epicatechin and (+)-catechin with [M − H]^−^ at *m*/*z* = 289. Moreover, the red apple ciders contained three B-type procyanidin dimers with *m*/*z* = 577 and one trimer with *m*/*z* = 865, which were identified using commercial standards and literature [27,28,29,31] in apple fruits and ciders. The yeast strain used in the fermentation process had a significant (*p* < 0.05) impact on flavan-3-ols content. The red apple ciders obtained with *S. cerevisiae* with the addition of BCD had a concentration of flavan-3-ols around 1.3 and 1.5 times higher compared to the ciders fermented with the addition of *S. cerevisiae* and *S. bayanus*, respectively. Significant differences were noted after storage and with the use of different yeast strains, especially *S. cerevisiae* with BCD, which caused a 5% increase in flavan-3-ols concentration in the ciders (*p* < 0.05), while the other yeast strains reduced their concentration after storage by 11% on average. These results confirm the protective effect of the BCD additive on the flavan-3-ol content. Flavan-3-ols are important components of wines and/or ciders. They are responsible for aroma loss, color of the finished product, and also oxidative browning. Flavan-3-ols, especially polymeric procyanidins, directly affect the taste of wines and ciders and contribute to bitterness and astringency development in products [34,35]. Moreover, polymeric procyanidins may undergo polymerization and oxidation, which may cause the browning of wines and ciders [36].

#### 2.2.3. Anthocyanins

The third fraction of the prepared red apple ciders turned out to be anthocyanins. They are responsible for the red to purple color of apple skin and products, and they also exhibit significant antioxidative properties. All seven identified anthocyanins were isomers of cyanidins. These compounds found in rose ciders or red wines often change from monomeric to polymeric, and these changes occur during vinification operations such as fermentation, storage, bottling, and aging. The major component of this fraction was detected as cyanidin-3-*O*-galactoside ([M − H]^−^ at *m*/*z* = 449) [31]. The polymerization reaction occurring in red wines affects the condensation of anthocyanin compounds. Therefore, a cyanidin-3-*O-*galactoside-(epi)catechin adduct ([M − H]^−^ at *m*/*z* = 765) was detected in red apple ciders. In addition, cyanidin-3-*O*-galactoside-4-vinyl-(epi)catechin ([M − H]^−^ at *m*/*z* = 761, Rt = 2.55 min), cyanidin-3-*O*-galactoside -(epi)catechin adduct ([M − H]^−^ at *m*/*z* = 737, Rt = 3.23min), cyanidin-3-*O*-galatcoside procyanidin B2 ([M − H]^−^ at *m*/*z* = 1025, Rt = 3.97 min), and 5-carboxypyranocyanidin-3-hexoside (gal) ([M − H]^−^ at *m*/*z* = 517, Rt = 4.91) were identified in red apple ciders after fermentation and storage. Another compound detected in ciders was cyanidin-3-*O*-pentoside with *m*/*z* = 419 and MS/MS at *m*/*z* = 287. These compounds were identified according to commercial standards, MS spectra, and literature data [28,32,37]. Significant (*p* > 0.05) differences were noted in the concentration of anthocyanins depending on yeast strain. The use of BCD increased their concentration in the ciders after storage 1.4 and 1.5 times compared to the products fermented with *S. cerevisiae* and *S. bayanus*. Similar results for the protection of anthocyanin compounds were obtained for chokeberry juice after clarification with BCD [9]. After storage, a ~16% increase was noted in anthocyanin concentration in the cider fermented with *S. cerevisiae* with the addition of BCD, while a little decrease in these compounds content was determined in other products. After aging time, anthocyanins concentration in the ciders ranged from 48.00 to 91.70 mg/L. The red apple ciders fermented with *S. cerevisiae* SIHAFERM Finesse Red with the addition of BCD had the highest content of color compounds. The results obtained confirm the protective effect of these yeasts with or without addition of BCDs on anthocyanins. In this group, the predominant anthocyanin was cyanidin-3-*O*-galactoside, which accounted for 70% of all these components, which is in agreement with findings reported by Knebel et al. [38] for red apple juice. Furthermore, red-colored proanthocyanins are formed during the processing and maturation of fruit juices [38]. In turn, carboxypyranoanthocyanin is formed under the influence of a cycloaddition of pyruvic acid in the enol form and cyanidin-3-*O*-galactoside. Then, the color of this compound is more resistant to pH changes. Moreover, changes that take place during ciders production and storage time result in browning, which not only negatively affects sensory properties of the product but also decreases its antioxidative capacity [38,39]. Therefore, it is important to choose the right yeast for the production of red apple cider and to control the changes during storage which could affect these compounds.

#### 2.2.4. Dihydrochalcones

Another group of identified phenolic compounds was dihydrochalcones, determined as four compounds with two derivatives of phloretin (with a fragment at *m*/*z* = 289) and two derivatives of phlorizin (with MS/MS at *m*/*z* = 273). The major components of this fraction were represented by free phlorizin ([M − H]^−^ at *m*/*z* = 435) and phloretin-2′-*O*-(2′′-*O*-xylosylglucoside) (with *m*/*z* = 567). In addition, two derivatives of phloretin were identified as hydroxyphloretin diglycoside with *m*/*z* = 583 and hydroxyphloretin monoglycoside with *m*/*z* = 451. These compounds are typical of apples and their products [9,27,28,29]. There was no significant (*p* > 0.05) effect of yeast strain and aging time on the content of this fraction of phenolics. After storage, dihydrochalcone concentration in red apple ciders ranged from 35.87 to 45.88 mg/L. The ciders fermented with Lalvin C (*S. bayanus*) had the lowest content of dihydrochalcones. In addition, the major compound in this group was phlorizin, which accounted for 45% of all dihydrochalcones, and this was in agreement with findings reported by other authors [10,28] for green apples. In addition, Alberti et al. [13] reported that the content of the analyzed compounds determined in juice and cider obtained from green apples was 2.1 and 2.4 times lower compared to the ciders produced from red apples. Dihydrochalcone, mainly phloretin, enhances the action of active components on the surface of lipids, and exhibits strong antioxidative properties [13].

#### 2.2.5. Flavonols 

The last fraction of phenolics was represented by five quercetins and their derivatives (with a fragment at *m*/*z* = 301) belonging to flavonols. The moieties were determined by classifying them as glucoside (loss of 162 Da from the molecular ion), rhamnoside (loss of 146 Da from the molecular ion), and pentoside (loss of 162 Da from the molecular ion). They were all determined by other authors in green apples [31]. The tested variants of ciders differed significantly (*p* > 0.05) in this respect depending on yeast strain used and BCD addition. However, yeast strains, especially *S. cerevisiae* with BCD, caused a 1.4- and 1.6-fold increase of flavonols content in ciders (*p* < 0.05) compared to the products fermented using *S. cerevisiae* and *S. bayanus* alone, respectively. In addition, Alberti et al. [13] reported that the flavonol content determined in green apple juice and cider was 1.7 times higher than in the ciders obtained from red apples. These compounds were the least abundant fraction of the compounds analyzed in red apple ciders; nevertheless, they can exhibit influence the final sensory quality of ciders because they are characterized by low thresholds of astringency. Moreover, flavonols, especially quercetin derivatives, can present anti-inflammatory, anti-allergic, anticoagulant, antitumor, and antiviral properties [31,40]. 

### 2.3. Determination of Antioxidative Capacity in Red Apple Cider

The antioxidative capacity of the red apple ciders was evaluated based on free radical-scavenging ability (ABTS) and ferric reducing antioxidative power (FRAP) assays (Table 5). Analyses of all samples before and after storage provided the same trends. No significant (*p* > 0.05) differences were found between the analyzed variants of samples as affected by yeast strain, storage time, and BCD addition. The antioxidative potential of red apple ciders assessed with the ABTS method ranged from 2.20 and 2.50 mmol Trolox equivalent (TE)/L after fermentation, and from 2.10 to 2.50 mmol TE/L after storage at 4 °C. The antioxidative potential of red apple ciders after aging at 4 °C ranged from 2.30 to 2.60 mmol TE/L (FRAP method). The antioxidative activity tested in green apple juice and ciders with the FRAP method was 2.3 and 2.5 times lower than the average FRAP value measured in red apple ciders [13]. In addition, the major fraction of phenolic compounds presented the highest significant correlation with the antioxidative potential of the analyzed ciders (ABTS, *r*^2^ = 0.96; FRAP, *r*^2^ = 0.83; *p* < 0.05), which was probably due to the presence of procyanidin polymers, which are known for their high antioxidative potency. Moreover, contents of anthocyanins, phenolic acids, and organic acids were positively correlated with the antioxidative capacity of red apple ciders (ABTS, *r*^2^ = 0.64; FRAP, *r*^2^ = 0.55; ABTS, *r*^2^ = 0.49; FRAP, *r*^2^ = 0.34; ABTS, *r*^2^ = 62; FRAP, *r*^2^ = 0.49; *p* < 0.05). The high correlation between the biologically active compounds and antioxidative capacity was confirmed in studies of juices and ciders from green apples carried out by Alberti et al. [13] and Laaksonen et al. [31].

### 2.4. Consumer Evaluation of Red Apple Ciders

The outcome of sensory evaluation of red apple ciders considering their taste, aroma, color, and consistency is illustrated in Figure 2. Generally, the observation after consumer evaluation showed that all red apple ciders, after aging for three months with the addition of yeast (*S. bayanus* (Lalvin QA23) and *S. cerevisiae* (Pure Nature, Rubino Cru, White Arome, SIHAFERM Finesse Red)), were attractive in terms of aroma at ≥4.5 units. In turn, red apple cider after fermentation received the lowest score for consistency at ≥3.1 units. The most desirable taste turned out to be that of the ciders fermented with the addition of *S. cerevisiae*: Rubino Cru (around 4.8 units) and SIHAFERM Finesse Red (around 4.5 units). According to the consumers, the least acceptable in terms of taste after storage was red apple cider fermented with *S. bayanus* Lalvin C yeast (around 3.1 units). After the sensory evaluation of taste, aroma, consistency, and color, the best cider produced from red apples turned out to be that fermented by *S. cerevisiae* Rubino Cru and SIHAFERM Finesse Red. It was probably associated with a lesser degree of polymerization (DP) and, hence, lower bitterness. In turn, Nurgel et al. [41] stated that the taste and color are important distinguishing factors taken into account in the consumer assessment of wine. The use of yeasts influenced the protection of an attractive red color, and the protection of compounds with health-promoting effects.

### 2.5. Principal Component Analysis (PCA)

Results of the PCA revealed correlations between organic acid content, values of basic physicochemical parameters, polyphenolic compound content, and antioxidative activity of stored new red apple cider fermented with various yeast strains (Figure 3). The two principal components (PCs) explained 72.60% of the total variance of tested data, including 51.58% reported for PC1 and 21.06% for PCPC1 was primarily accountable for the disparity between contents of anthocyanins, flavonols, phenolic acids, flavan-3-ols, dihydrochalcones, polymeric procyanidin, organic acids (mainly oxalic and fumaric acids), and antioxidative activity determined with the ABTS assay. In turn, PC2 coupled antioxidative potency with total acidity, as well as contents of organic acids and alcohol. This statistical analysis proved differences between the cider samples as affected by yeast strain and BCD addition during fermentation. The highest concentration of phenolic acids of all groups and of oxalic and fumaric acids in red apple ciders fermented with *S. cerevisiae* SIHAFERM Finesse Red with the addition of BCD described before depicted a strong positive correlation with the antioxidative potency assessed mainly with the ABTS test. Moreover, the high value of the color parameters a* and b* was responsible for the high proportion of red and yellow colors in these red apple ciders. The use of yeasts confirmed the protective effect of the color of the final product before and after storage, as well as the content of polyphenolic compounds with pro-health effects. Furthermore, high concentrations of organic acids (mainly malic, quinic, and succinic acids) and of alcohol in red apple ciders fermented with *S. cerevisiae* White Aroma and *S. bayanus* Lalvin QA23 and Lalvin C were strongly correlated with FRAP capacity. In contrast, in red apple ciders fermented with *S. cerevisiae* Pure Nature and Rubin Cru, the low content of phenolic compounds and high content of citric, acetic, and maleic acids, as well as the sum of these acids, demonstrated a positive correlation with contents of biologically active compounds and antioxidative properties. Additionally, these ciders were characterized by a high value of parameter L*, which means that they were the brightest. Alberti et al. [13] used PCA analysis to evaluate all variables in green apple beverages and ciders. They reported differences between the tested components and products. There were much more volatile compounds in ciders than amino acids, while an opposite trend was observed in beverages. In addition, the results showed that apple cultivar had a significant impact on the chemical composition of the final ciders made from green apples.

## 3. Materials and Methods 

### 3.1. Reagent and Standard 

The compounds 2,2′-Azinobis(3-ethylbenzothiazoline-6-sulfonic acid) (ABTS), 6-hydroxy-2,5,7,8-tetramethylchroman-2-carboxylic acid (Trolox), 2,4,6-tri(2-pyridyl)-*s*-triazine (TPTZ), methanol acetic acid, phloroglucinol, *β*-cyclodextrin, and K_2_S_2_O_5_ were purchased from Sigma-Aldrich (Steinheim, Germany). On the other hand, (−)-epicatechin, (+)-catechin, procyanidin B2, chlorogenic acid, neochlorogenic acid, cryptochlorogenic acid, caffeic acid, dicaffeoylquinic acid, *p*-coumaric acid, myricetin, isoquercitrin, cyanidin-3-*O*-galactoside, and cyanidin-3-*O*-glucoside were purchased from Extrasynthese (Lyon, France). Acetonitrile for ultra-performance liquid chromatography (UPLC; Gradient grade) and ascorbic acid were from Merck (Darmstadt, Germany). Yeasts *S. cerevisiae* and *S. bayanus* were purchased from Eaton company (Sandomierz, Poland).

### 3.2. Cider Production 

The study involved the red apple of the “Bella Marii” cultivar. The material was collected from a Grzegorz Maryniowski BioGrim company in Wojciechów (51°10′22″ north (N), 23°03′27″ east (E), Poland), near Lublin (Poland), in the harvested red apples were fully mature. 

The red apples were crushed with a Thermomix (Vorwerk, Wuppertal, Germany). The crushed bulk was pressed with a hydraulic press (TOYA, Wrocław, Poland). The must was supplemented with K_2_S_2_O_5_ (POCh Gliwice, Poland) at 0.10 g/L, nutrients (SIHA Proferm Plus) at 0.10 g/L (Eaton, Begerow, Langenlonsheim, Germany), and *S. cerevisiae* (SIHAFERM Pure Nature, SIHA Rubino Cru, SIHA White Arome, SIHAFERM Finesse Red) and *S. bayanus* (Lalvin C, Lalvin QA 23 YSEO) at 0.20 g/L. Sample 7 was without potassium metabisulfite, but with BCD added at 1.0 g/L with SIHAFERM Finesse Red at 0.20 g/L. The fermentation was conducted at 20 °C for 15 days. Afterward, the red apple cider was stored at 4 °C for three months. 

### 3.3. Physical Analysis 

Titratable acidity and pH were determined by titration aliquots (Schott Titroline 7500 KF Volumetric KF Titrator; Mainz, Germany), performed according to the protocol described by Lachowicz et al. [42]. Turbidity was measured using a method described previously by Lachowicz et al. [42]. Ethanol content in wine was determined using an oscillating densimeter DMA 4500 M (Anton Paar, Graz, Austria), with results presented as the volume percentage (vol.%). Results are reported as the arithmetic mean of three independent repetitions, taking into account the standard deviation (SD).

### 3.4. Qualitative and Quantitative Assessment of Polyphenols 

All analyses of polyphenols in the tested samples were carried out using an ACQUITY Ultra-Performance LC system (UPLC) equipped with a binary solvent manager (Waters Corp., Milford, MA, USA), a UPLC ethylene-bridged hybrid (BEH) C18 column (1.7 μm, 2.1 mm × 50 mm, Waters Corp., Milford, MA, USA), and a QToF Micro mass spectrometer (Waters, Manchester, UK) with an electrospray ionization (ESI) source operating in negative and positive modes. The analysis was carried out using full-scan, data-dependent MS scanning from *m*/*z* 100 to Leucine enkephalin was used as the reference compound at a concentration of 500 pg/μL, and a flow rate of 2 μL/min, and the [M − H]^−^ ion at 554.2615 Da was detected. The [M − H]^−^ ion was detected during 15-min analysis performed within ESI MS accurate mass experiments, which were permanently introduced via the LockSpray channel using a Hamilton pump. The lock mass correction was ± 1.000 for the mass window. The mass spectrometer was operated in the negative-ion mode, set to the base peak intensity (BPI) chromatograms, and scaled to 12,400 counts per second (cps) (100%). The optimized MS conditions were as follows: capillary voltage of 2500 V, cone voltage of 30 V, source temperature of 100 °C, dissolution temperature of 300 °C, and dissolution gas (nitrogen) flow rate of 300 L/h. Collision-induced fragmentation experiments were performed using argon as the collision gas, with voltage ramping cycles from 0.3 to 2 V. The data obtained from UPLC–MS were subsequently entered into the MassLynx 4.0 ChromaLynx Application Manager software. A protocol described earlier by Lachowicz et al. [42] was followed during the extraction and determination of phenolic compounds. The mobile phase consisted of solvent A (4.5% formic acid, *v*/*v*) and solvent B (100% acetonitrile). The runs were monitored at the following wavelengths: phenolic acids at 320 nm, flavonols at 340 nm, anthocyanins at 520 nm, and flavan-3-ols at 280 nm. The photodiode array detector (PDA) spectra were measured over the wavelength range of 200–600 nm in steps of 2 nm. The results were expressed as mg/L.

### 3.5. Analysis of Proanthocyanidins via Phloroglucinolysis Method 

Phloroglucinolysis products were separated on a Cadenza CD-C18 (75 mm × 4.6 mm, 3 μm) column (Imtakt, Kyoto, Japan). The liquid chromatograph was a Waters (Milford, MA, USA) system equipped with diode array, scanning fluorescence detectors (Waters 474), and an autosampler (Waters 717 plus). Solvent A (25 mL of acetic acid and 975 mL of water) and solvent B (acetonitrile) were used in the following gradients: initial, 5 mL/100 mL B; 0–15 min, to 10 mL/100 mL B linear; 15–25 min to 60 mL/100 mL B linear; followed by washing and reconditioning of the column. A flow rate of 1 mL/min and an oven temperature of 15 °C were used with the injection of the filtrate (20 μL) into the HPLC system. The fluorescence detection was recorded at an excitation wavelength of 278 nm and an emission wavelength of 360 nm. The calibration curves were established using (+)-catechin and (−)-epicatechin (phloroglucinol adduct standards). All data were obtained in triplicate. The results were expressed as mg/L.

### 3.6. Determination of Antioxidative Activity

The free-radical scavenging activities were determined using two methods, ABTS and FRAP (ferric reducing antioxidative power). The ABTS and FRAP assays were conducted as previously described by Re et al. [43] and Benzie and Strain [44], respectively. Determinations by ABTS and FRAP methods were performed using a UV-2401 PC spectrophotometer (Shimadzu, Kyoto, Japan). The antioxidative activity was evaluated by measuring the variation in absorbance at 734 nm after 6 min for ABTS, and at 593 nm after 10 min for FRAP. All antioxidative activity analyses were done in triplicate, and results were expressed as mmol of Trolox equivalent (TE) per L of sample.

### 3.7. Organic Acids

Organic acids were determined by HPLC–PDA as described previously by Wojdyło et al. [45]. All data were obtained in triplicate. Results were expressed as mg/L of red apple cider.

### 3.8. Color Measured

Color properties (L*, a*, b*) of cider from red apple were determined by reflectance measurements with a Color Quest XE Hunter Lab colorimeter. The samples were determined according to the method described by Lachowicz et al. [42]. The data were means of three measurements. The total change in cider before and after storage was expressed as ΔE, and the chroma difference (ΔC) [42] was also calculated.

### 3.9. Sensory Attributes

The sensory properties of obtained red apple cider before and after three months of storage at 4 °C were evaluated using a five-degree hedonic scale with boundary indications: “I do not like it very much” (1) to “I like it very much” (5). The assessment included the following quality attributes: taste, aroma, color, and consistency. It was conducted by a group of 20 consumer panelists (10 men and 10 women in the age group of 20–65). Coded samples were provided to the panelists for the evaluation at 20 °C in uniform 50-mL plastic containers.

### 3.10. Statistical Analysis

Statistical analysis was conducted using Statistica version 12.5 (StatSoft, Krakow, Poland). Significant differences (*p* < 0.05) between means were evaluated by one-way ANOVA, PCA (principal components analysis), and Duncan’s multiple range test. All analyses were done in triplicate.

## 4. Conclusions 

This work provides information on the impact of different yeast strains, addition of BCD, and storage time on the physicochemical properties, antioxidative potential, and final quality of red apple ciders.

Seven variants of red apple ciders were obtained, depending on the yeast strain (*S. cerevisiae* and *S. bayanus*), BCD addition, and aging time. Generally, storage at 4 °C had an effect on the quality of the tested components of red apple ciders. Yeast strain and addition of BCD influenced their variety. *S. cerevisiae*, especially SIHAFERM Finesse Red with BCD and SIHAFERM Finesse Red strain, allowed producing a red apple cider of better physicochemical properties and antioxidative potential than that produced with *S. bayanus*. SIHAFERM Finesse Red yeast of the new generation influenced the stability and ensured the protection of color and polyphenols, especially anthocyanins, therefore they are recommended for the fermentation of red apple ciders. The addition of BCD affected the protection of polyphenolic compounds, which exhibit the health-promoting properties, and also influenced basic chemical parameters, organic acids, and color of the ciders, nevertheless the taste of these ciders was unacceptable.

## Figures and Tables

**Figure 1 molecules-24-02477-f001:**
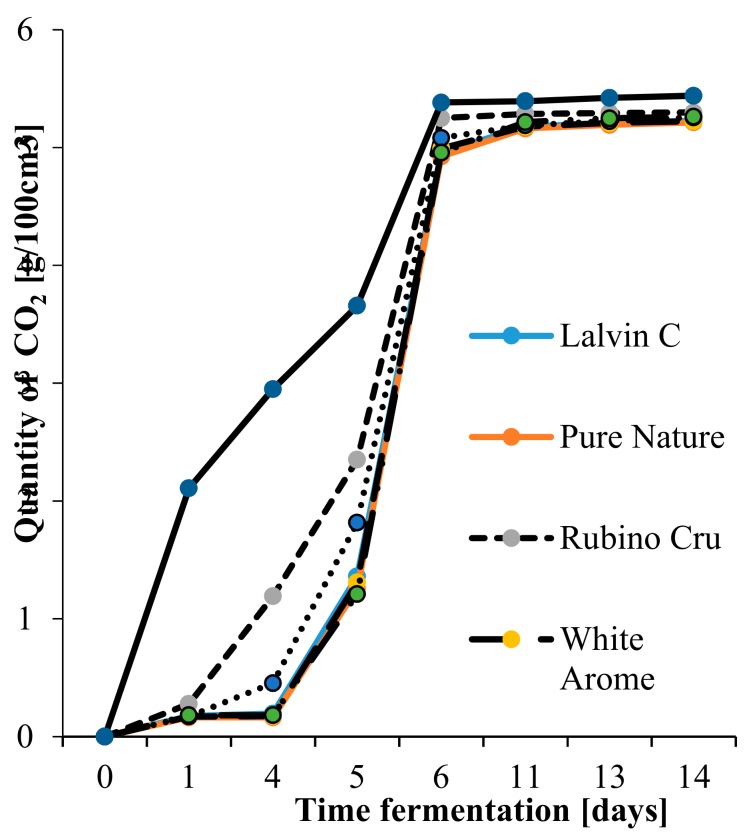
Dynamic fermentation in red apple ciders.

**Figure 2 molecules-24-02477-f002:**
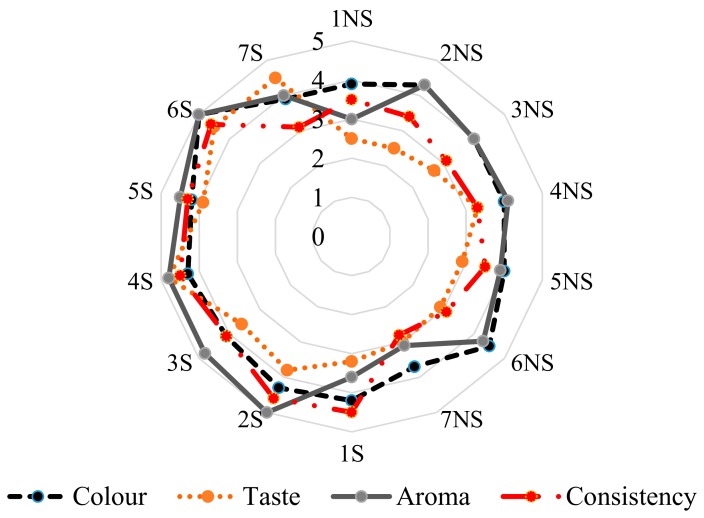
Sensory evaluation in red apple ciders before and after storage. NS, after fermentation; S, after storage time.

**Figure 3 molecules-24-02477-f003:**
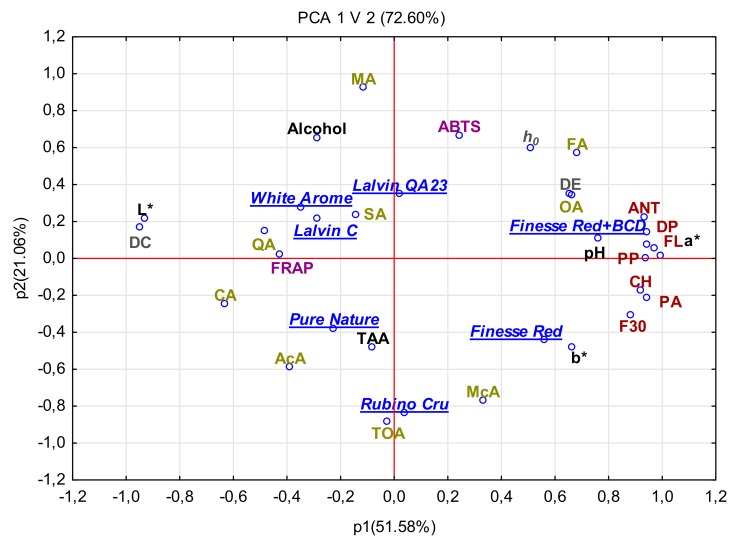
Principal component analysis (PCA) in red apple ciders after storage. ANT, anthocyanins; PA, phenolic acid; PP, polymeric procyanidin; CH, chalcones; F3O, flavonols (monomers and oligomers); DP, degree of polymerization; FL, flavonols; TOA, sum of organic acids; CA, citric acid; AcA, acetic acid; McA, maleonic acid; QA, quinic acid; OA, oxalic acid; SA, succinic acid; FA, fumaric acid; MA, malic acid.

**Table 1 molecules-24-02477-t001:** The alcohol, total acidity, pH values, and color parameters in red apple ciders. L*—lightness; a*—redness; b*—yellowness; ΔE—total color difference; ΔC—chroma difference; BCD—*β*-cyclodextrin.

Storage	Type of Yeast	Product	Alcohol (% *v*/*v*)	Total Acidity (g/L)	pH	Color Parameters				
						L*	a*	b*	Δ *E*	Δ *C*
	-	Juice				65	45	53	-	-
Before storage	*Saccharomyces* *bayanus*	Lalvin C	6.92 ± 0.01 ^2^	6.48 ± 0.01	3.25 ± 0.00	61.78 ± 0.07	32.35 ± 0.04	28.27 ± 0.03	27.96	26.57
		Lalvin QA23	6.91 ± 0.01	5.95 ± 0.01	3.29 ± 0.00	54.61 ± 0.07	33.73 ± 0.04	35.46 ± 0.04	23.29	20.59
	*Saccharomyces* *cerevisiae*	Pure Nature	6.92 ± 0.01	6.20 ± 0.01	3.22 ± 0.00	59.56 ± 0.07	32.48 ± 0.04	30.81 ± 0.04	26.05	24.76
		Rubino Cru	6.90 ± 0.01	6.33 ± 0.01	3.27 ± 0.00	39.75 ± 0.05	34.22 ± 0.04	42.70 ± 0.05	29.32	14.81
		White Arome	6.90 ± 0.01	6.1 ± 0.01	3.28 ± 0.00	58.85 ± 0.07	32.93 ± 0.04	31.80 ± 0.04	25.16	23.75
		SIHAFERM Finesse Red	6.91 ± 0.01	6.21 ± 0.01	3.28 ± 0.00	52.75 ± 0.06	37.77 ± 0.05	37.07 ± 0.04	21.36	16.60
		SIHAFERM Finesse Red + BCD	6.88 ± 0.01	6.49 ± 0.01	3.27 ± 0.00	31.08 ± 0.04	42.90 ± 0.05	44.94 ± 0.05	34.93	7.40
After 3 months of storage	*Saccharomyces* *bayanus*	Lalvin C	6.88 ± 0.01	6.19 ± 0.01	3.29 ± 0.00	67.31 ± 0.08	34.45 ± 0.04	19.74 ± 0.02	34.97	29.82
		Lalvin QA23	6.87 ± 0.01	6.71 ± 0.01	3.28 ± 0.00	64.41 ± 0.08	38.94 ± 0.05	21.64 ± 0.03	31.95	24.98
	*Saccharomyces* *cerevisiae*	Pure Nature	6.85 ± 0.01	6.88 ± 0.01	3.26 ± 0.00	66.54 ± 0.08	34.77 ± 0.04	20.30 ± 0.02	34.30	29.26
		Rubino Cru	6.85 ± 0.01	6.62 ± 0.01	3.29 ± 0.00	49.13 ± 0.06	39.22 ± 0.05	30.47 ± 0.04	28.16	19.86
		White Arome	6.92 ± 0.01	6.47 ± 0.01	3.25 ± 0.00	63.6 ± 0.08	34.51 ± 0.04	23.00 ± 0.03	31.81	28.05
		SIHAFERM Finesse Red	6.92 ± 0.01	6.29 ± 0.01	3.32 ± 0.00	61.89 ± 0.07	37.01 ± 0.04	23.87 ± 0.03	30.37	25.49
		SIHAFERM Finesse Red + BCD	6.87 ± 0.01	6.39 ± 0.01	3.34 ± 0.00	32.38 ± 0.04	49.57 ± 0.06	29.09 ± 0.03	40.70	12.05
Type of yeast	*Saccharomyces cerevisiae*		6.89 a ^1^	6.40 b	3.28 b	51.55 b	37.54 b	31.41 b	30.22 b	20.20 b
	*Saccharomyces cerevisiae* + BCD		6.88 a	6.44 a	3.31 a	31.73 c	46.24 a	37.02 a	37.82 a	9.73 c
	*Saccharomyces bayanus*		6.90 a	6.33 c	3.28 b	62.03 a	34.87 c	26.28 c	29.54 c	25.49 a
Storage time	Before storage		6.91 A	6.25 B	3.27 A	51.20 B	35.20 B	35.86 A	26.87 B	19.21 B
	After 3 months of storage		6.88 B	6.51 A	3.29 A	57.89 A	38.35 A	24.02 B	33.18 A	24.22 A

^1^ a–c; A–B: means ± SD followed by different letters within the same line represent significant differences (*p* < 0.05). Data are the averages of triplicates. ^2^ Values are means ± standard deviation; *n* = 3.

**Table 2 molecules-24-02477-t002:** The content of organic acid in red apple ciders**.**

	Type of Yeast	Product	Organic Acid (mg/L)								
			Oxalic	Maleic	Citric	Malic	Quinic	Succinic	Fumaric	Acetic	SUM
Before storage	*Saccharomyces* *bayanus*	Lalvin C	353.12 ± 0.35	9.97 ± 0.01	1389.85 ± 1.67	5606.12 ± 1.12	376.66 ± 0.08	211.81 ± 0.04	7.24 ± 0.01	6.47 ± 0.01	7961.24
		Lalvin QA23	337.34 ± 0.34	10.24 ± 0.01	1356.18 ± 1.63	5444.10 ± 1.09	308.03 ± 0.06	256.31 ± 0.05	7.38 ± 0.01	6.85 ± 0.01	7726.43
	*Saccharomyces* *cerevisiae*	Pure Nature	334.02 ± 0.33 ∞	11.59 ± 0.01	1312.81 ± 1.58	5670.14 ± 1.13	335.40 ± 0.07	219.67 ± 0.04	7.84 ± 0.01	5.55 ± 0.01	7897.02
		Rubino Cru	342.04 ± 0.34	11.28 ± 0.01	1309.43 ± 1.57	5574.53 ± 1.11	323.47 ± 0.06	249.90 ± 0.05	9.02 ± 0.01	6.46 ± 0.01	7826.13
		White Arome	337.66 ± 0.34	11.5 ± 0.01	1293.74 ± 1.55	5661.29 ± 1.13	318.41 ± 0.06	217.96 ± 0.04	8.3 ± 0.01	5.67 ± 0.01	7854.53
		SIHAFERM Finesse Red	329.69 ± 0.33	10.07 ± 0.01	1382.83 ± 1.66	5090.53 ± 1.02	301.32 ± 0.06	224.62 ± 0.04	6.83 ± 0.01	4.77 ± 0.01	7350.66
		SIHAFERM Finesse Red + BCD	331.28 ± 0.33	12.56 ± 0.02	1250.98 ± 1.5	5689.25 ± 1.14	298.16 ± 0.06	193.69 ± 0.04	7.16 ± 0.01	4.93 ± 0.01	7788.01
After 3 months of storage	*Saccharomyces* *bayanus*	Lalvin C	322.04 ± 0.32	11.31 ± 0.01	1162.95 ± 1.4	5740.54 ± 1.15	347.63 ± 0.07	216.25 ± 0.04	7.59 ± 0.01	5.27 ± 0.01	7813.58
		Lalvin QA23	406.08 ± 0.41	12.32 ± 0.01	1026.88 ± 1.23	5894.11 ± 1.18	338.61 ± 0.07	247.80 ± 0.05	14.49 ± 0.02	4.36 ± 0.01	7944.65
	*Saccharomyces* *cerevisiae*	Pure Nature	333.47 ± 0.33	10.92 ± 0.01	1269.91 ± 1.52	5948.37 ± 1.19	323.15 ± 0.06	244.21 ± 0.05	6.16 ± 0.01	5.53 ± 0.01	8141.72
		Rubino Cru	331.46 ± 0.33	10.14 ± 0.01	1224.22 ± 1.47	6023.83 ± 1.2	321.98 ± 0.06	242.97 ± 0.05	6.08 ± 0.01	5.51 ± 0.01	8166.19
		White Arome	338.24 ± 0.34	12.06 ± 0.01	1291.27 ± 1.55	5646.26 ± 1.13	322.56 ± 0.06	212.35 ± 0.04	8.14 ± 0.01	5.52 ± 0.01	7836.40
		SIHAFERM Finesse Red	322.91 ± 0.32	11.69 ± 0.01	1334.20 ± 1.60	5736.63 ± 1.15	307.97 ± 0.06	214.91 ± 0.04	8.19 ± 0.01	4.78 ± 0.01	7941.28
		SIHAFERM Finesse Red + BCD	393.81 ± 0.39	11.34 ± 0.01	1051.58 ± 1.26	5864.26 ± 1.17	307.91 ± 0.06	245.48 ± 0.05	14.37 ± 0.02	4.86 ± 0.01	7893.61
Type of yeast	*Saccharomyces cerevisiae*		339.46 c ^1^	11.32 b	1272.10 b	5690.51 b	316.03 b	226.58 b	8.21 c	5.36 b	7869.56 a
	*Saccharomyces cerevisiae* + BCD		362.55 a	11.95 a	1151.28 c	5776.76 a	303.04 c	219.59 c	10.77 a	4.90 c	7840.81 c
	*Saccharomyces bayanus*		354.65 b	10.96 c	1233.97 a	5671.22 c	342.73 a	233.04 a	9.18 b	5.74 a	7861.48 b
Storage time	Before storage		337.88 B	11.03 B	1327.97 A	5533.71 B	323.06 B	224.85 A	7.68 B	5.81 A	7772.00 B
	After storage		349.72 A	11.40 A	1194.43 B	5836.29 A	324.26 A	232.00 B	9.29 A	5.12 B	7962.49 A

^1^ a–c; A–B: means ± SD followed by different letters within the same line represent significant differences (*p* < 0.05). Data are the averages of triplicates. ^2^ Values are means ± standard deviation; *n* = 3.

**Table 3 molecules-24-02477-t003:** Identification of polyphenols in red apple ciders.

No	Tentative Identification^†^	Retention Time [min]	Molecular Ion MS [H − M]^−^	Fragments MS/MS (*m*/*z*)
1	Cyanidin-3-galactoside -4-vinyl-(epi)catechine	2.55	761	287
24	(+)-Catechin	2.65	289
2	Cyanidin-3-galactoside -(epi)catechine adduct	3.23	737	287
8	3-*O-p*-coumaroylquinic acid	3.28	337	163
25	Procyanidin B2	3.55	577	289
3	Cyanidin-3-*O*-galactoside -(epi)catechine adduct with acetaldehyde	3.64	765	287
26	Procyanidin B2	3.74	577	289
9	3-*O*-caffeoylquinic acid	3.83	353	191
4	Cyanidin-3-galatcoside dimmer procyanidin	3.97	1025	287
27	(-)-Epicatechin	4.08	289	
10	5-*O*-caffeoylquinic acid	4.16	353	191
11	4-*O*-caffeoylquinic acid	4.30	353	191
5	Cyanidin-3-galactoside	4.45	449	287
28	Procyanidin B2	4.62	577	289
12	4-*O-p*-coumaroylquinic acid	4.81	337	173
6	5-Carboxypyranocyanidin-3-hexoside (gal)	4.91	517	355
13	Caffeic acid	4.96	179	135
14	5-*O-p-*coumaroylquinic acid	5.12	337	191
29	Procyanidin B3	5.38	865	289
7	Cyanidin pentoside	5.47	419	287
15	Ethyl chlorogenate mcz	5.64	382	191
16	Ethyl 3-*O-p*-coumaroylquinic acid	6.09	365	163
17	Ethyl 3-*O-p*-coumaroylquinic acid	6.33	365	163
18	Caffeic acid	6.40	179	
30	Hydroxyphloretin diglycoside	6.65	583	451/289
19	Quercetin-*3-O*-galactoside	6.78	463	301
20	Quercetin-*3-O*-glucoside	6.90	463	301
21	Quercetin-*3-O*-pentoside	7.19	433	301
31	Hydroxyphloretin monoglycosid	7.32	451	289
32	Phloridzin	7.50	435	273
22	Quercetin-*3-O*-rhamnoside	7.67	447	301
33	Phloretin-2′-*O*-(2″-*O*-xylosylglucoside)	8.24	567	273
23	Quercetin	9.86	301	

**Table 4 molecules-24-02477-t004:** The average content of polyphenolic compounds in red apple ciders before and after storage (mg/L).

Group	Compounds	Before Storage							Type of Yeast			Storage Time	
		*S. bayanus*		*S. cerevisione*									
		C1^6^	C6	C2	C3	C4	C5	C7	*S. cerevisione*	*S. cerevisione**+* B*CD*	*S. bayanus*	Om	3m;4 °C
Anthocyanins	1	1.04 ± 0.01	1.46 ± 0.01	1.77 ± 0.01	1.47 ± 0.01	1.91 ± 0.01	1.78 ± 0.01	1.91 ± 0.01	1.62a^1^	1.04b	1.61a	1.63A	1.61A
	2	3.13 ± 0.02	4.01 ± 0.02	3.90 ± 0.02	4.00 ± 0.02	4.15 ± 0.02	3.78 ± 0.02	4.26 ± 0.03	3.86b	3.13c	3.95a	3.96A	3.81A
	3	0.81 ± 0.00	1.04 ± 0.01	0.91 ± 0.01	0.87 ± 0.01	1.03 ± 0.01	1.02 ± 0.01	1.08 ± 0.01	0.96a	0.81b	0.97a	0.98A	0.95A
	4	0.29 ± 0.00	1.93 ± 0.01	2.28 ± 0.01	1.80 ± 0.01	2.37 ± 0.01	2.13 ± 0.01	2.10 ± 0.01	1.74b	0.29c	2.11a	1.97A	1.71B
	5	70.22 ± 0.42	37.54 ± 0.23	38.68 ± 0.23	39.05 ± 0.23	31.96 ± 0.19	35.53 ± 0.21	39.43 ± 0.24	43.24b	70.22a	38.11c	41.42B	42.12A
	6	3.65 ± 0.02	3.53 ± 0.02	4.12 ± 0.02	3.50 ± 0.02	3.83 ± 0.02	3.79 ± 0.02	3.97 ± 0.02	3.75b	3.65c	3.82a	3.88A	3.65B
	7	5.04 ± 0.03	3.92 ± 0.02	4.22 ± 0.03	3.9 ± 0.02	3.23 ± 0.02	3.57 ± 0.02	3.44 ± 0.02	3.83c	5.04a	4.07b	3.96A	3.84B
	SUM	84.17 ± 0.5	53.4 ± 0.32	55.87 ± 0.34	54.57 ± 0.33	48.46 ± 0.29	51.6 ± 0.31	56.16 ± 0.34	58.99b	84.17a	54.63c	57.79A	57.69A
Phenolic acids	8	1.85 ± 0.01	1.59 ± 0.01	1.59 ± 0.01	1.64 ± 0.01	1.77 ± 0.01	1.59 ± 0.01	1.68 ± 0.01	1.70b	1.85a	1.59c	1.69A	1.65A
	9	15.04 ± 0.09	14.3 ± 0.09	14.32 ± 0.09	14.66 ± 0.09	15.28 ± 0.09	13.87 ± 0.08	15.15 ± 0.09	14.80b	15.04a	14.31c	15.16A	14.15B
	10	133.49 ± 0.80	113.23 ± 0.68	113.89 ± 0.68	114.14 ± 0.68	119.7 ± 0.72	111.45 ± 0.67	116.59 ± 0.70	119.07c	133.49a	113.56b	122.75A	112.24B
	11	3.67 ± 0.02	1.72 ± 0.01	2.06 ± 0.01	1.64 ± 0.01	1.68 ± 0.01	1.82 ± 0.01	2.02 ± 0.01	2.16b	3.67a	1.89c	1.94B	2.22A
	12	3.54 ± 0.02	3.16 ± 0.02	3.09 ± 0.02	3.22 ± 0.02	3.39 ± 0.02	3.22 ± 0.02	3.25 ± 0.02	3.32b	3.54a	3.12c	3.47A	3.06B
	12	6.75 ± 0.04	6.79 ± 0.04	6.62 ± 0.04	6.88 ± 0.04	7.32 ± 0.04	6.90 ± 0.04	6.95 ± 0.04	6.96a	6.75b	6.70c	7.27A	6.49B
	14	5.72 ± 0.03	5.17 ± 0.03	5.12 ± 0.03	5.23 ± 0.03	5.53 ± 0.03	5.15 ± 0.03	5.28 ± 0.03	5.38b	5.72a	5.15c	5.58A	5.05B
	15	4.22 ± 0.03	2.42 ± 0.01	2.02 ± 0.01	2.01 ± 0.01	3.96 ± 0.02	3.17 ± 0.02	2.00 ± 0.01	3.07b	4.22a	2.22c	3.23A	2.42B
	16	0.40 ± 0.00	0.51 ± 0.00	0.52 ± 0.00	0.54 ± 0.00	0.66 ± 0.00	0.53 ± 0.00	0.57 ± 0.00	0.54a	0.40b	0.51a	0.49A	0.57A
	17	1.52 ± 0.01	0.94 ± 0.01	0.99 ± 0.01	0.94 ± 0.01	0.95 ± 0.01	0.99 ± 0.01	0.98 ± 0.01	1.07b	1.52a	0.96c	1.07A	1.01A
	18	1.67 ± 0.01	0.08 ± 0.00	0.23 ± 0.00	0.07 ± 0.00	1.75 ± 0.01	0.09 ± 0.00	0.10 ± 0.00	0.73b	1.67a	0.15c	0.49B	0.65A
	SUM	177.85 ± 1.07	149.9 ± 0.9	150.41 ± 0.9	150.95 ± 0.91	161.97 ± 0.97	148.75 ± 0.89	154.56 ± 0.93	158.81b	177.85a	150.15c	163.16A	149.52B
Flavonols	19	3.47 ± 0.02	2.51 ± 0.02	2.56 ± 0.02	2.50 ± 0.01	2.69 ± 0.02	2.61 ± 0.02	2.62 ± 0.02	2.78b	3.47a	2.53c	2.94A	2.47B
	20	1.01 ± 0.01	0.59 ± 0.00	0.60 ± 0.00	0.58 ± 0.00	0.60 ± 0.00	0.63 ± 0.00	0.60 ± 0.00	0.68b	1.01a	0.59c	0.75A	0.56B
	21	1.33 ± 0.01	0.85 ± 0.01	0.85 ± 0.01	0.84 ± 0.01	0.89 ± 0.01	0.89 ± 0.01	0.85 ± 0.01	0.96b	1.33a	0.85c	1.04A	0.82B
	22	4.66 ± 0.03	3.13 ± 0.02	3.23 ± 0.02	3.11 ± 0.02	3.38 ± 0.02	3.23 ± 0.02	3.33 ± 0.02	3.54b	4.66a	3.18c	3.52A	3.35A
	23	1.04 ± 0.01	0.03 ± 0.00	0.08 ± 0.00	0.03 ± 0.00	0.08 ± 0.00	0.07 ± 0.00	0.05 ± 0.00	0.25b	1.04a	0.05c	0.05B	0.34A
	SUM	11.51 ± 0.07	7.11 ± 0.04	7.32 ± 0.04	7.05 ± 0.04	7.62 ± 0.05	7.41 ± 0.04	7.44 ± 0.04	8.20b	11.51a	7.21c	8.30A	7.54B
Flavan-3-ols	24	3.83 ± 0.02	2.38 ± 0.01	3.53 ± 0.02	2.25 ± 0.01	3.23 ± 0.02	2.99 ± 0.02	2.95 ± 0.02	3.05b	3.83a	2.95c	3.25A	2.79B
	25	29.17 ± 0.17	26.64 ± 0.16	24.66 ± 0.15	24.67 ± 0.15	38.14 ± 0.23	27.59 ± 0.17	27.55 ± 0.17	29.42a	29.17b	25.65c	29.05A	27.64B
	26	12.05 ± 0.07	11.82 ± 0.07	11.55 ± 0.07	12.16 ± 0.07	11.66 ± 0.07	11.51 ± 0.07	12.01 ± 0.07	11.88b	12.05a	11.69c	12.10A	11.54B
	27	5.33 ± 0.03	5.35 ± 0.03	5.02 ± 0.03	5.26 ± 0.03	5.11 ± 0.03	5.34 ± 0.03	4.56 ± 0.03	5.12c	5.33a	5.19b	5.55A	4.73B
	28	13.44 ± 0.08	7.90 ± 0.05	8.15 ± 0.05	8.17 ± 0.05	8.71 ± 0.05	8.06 ± 0.05	7.99 ± 0.05	9.27b	13.44a	8.03c	10.41A	7.42B
	29	11.70 ± 0.07	3.09 ± 0.02	3.47 ± 0.02	3.13 ± 0.02	3.55 ± 0.02	3.28 ± 0.02	3.13 ± 0.02	4.96b	11.70a	3.28c	4.23B	4.72A
	SUM	75.5 ± 0.45	57.18 ± 0.34	56.37 ± 0.34	55.61 ± 0.33	70.38 ± 0.42	58.76 ± 0.35	58.18 ± 0.35	63.68b	75.50a	56.77c	64.58A	58.84B
	PP^3^	156.33 ± 0.94	102.38 ± 0.61	106.82 ± 0.64	113.41 ± 0.68	116.29 ± 0.70	109.12 ± 0.65	110.12 ± 0.66	121.05b	156.33a	104.60c	122.52A	110.18B
	DP^4^	9.95 ± 0.06	6.81 ± 0.04	6.92 ± 0.04	6.93 ± 0.04	7.63 ± 0.05	7.01 ± 0.04	7.32 ± 0.04	7.77b	9.95a	6.86c	8.14A	6.88B
Dihygrochalcones	30	6.65 ± 0.04	6.38 ± 0.04	6.46 ± 0.04	6.48 ± 0.04	6.81 ± 0.04	6.44 ± 0.04	6.97 ± 0.04	6.67a	6.65a	6.42a	6.81A	6.38B
	31	1.96 ± 0.01	1.89 ± 0.01	1.95 ± 0.01	1.90 ± 0.01	2.05 ± 0.01	1.92 ± 0.01	1.97 ± 0.01	1.96a	1.96a	1.92a	2.00A	1.90A
	32	19.60 ± 0.12	17.23 ± 0.10	17.55 ± 0.11	17.66 ± 0.11	18.15 ± 0.11	17.03 ± 0.10	18.21 ± 0.11	18.13b	19.60a	17.39c	18.44A	17.39B
	33	16.42 ± 0.10	13.05 ± 0.08	13.24 ± 0.08	13.14 ± 0.08	14.12 ± 0.08	13.04 ± 0.08	13.51 ± 0.08	14.04b	16.42a	13.15c	14.15A	13.42A
	SUM	44.62 ± 0.27	38.53 ± 0.23	39.19 ± 0.24	39.18 ± 0.24	41.13 ± 0.25	38.42 ± 0.23	40.66 ± 0.24	40.80b	44.62a	38.86c	41.39A	39.09B
TP^5^		562.57 ± 64.39	431.93 ± 54.15	436.14 ± 53.85	440.83 ± 55.40	463.76 ± 58.74	446.26 ± 55.65	448.00 ± 55.76	451.54b	549.96a	412.22c	457.75A	422.86B

^1^ a–c; A–B Means ± SD followed by different letters within the same line represent significant differences (*p* < 0.05). Data are the averages of triplicates; ^2^ Values are means ± standard deviation. *n* = 3; ^3^ PP, polymeric procyanidins; ^4^ DP, degree of polymerization; ^5^ TP, sum of phenolic compounds; ^6^ C1–C7, red apple dicers fermentation by Lalvin C, Lalvin QA 23 YSEO C, SIHAFERM Pure Nature, SIHA Rubino Cru, SIHA White Arome, SIHAFERM Finesse Red, and SIHAFERM Finesse Red + BCD.

**Table 5 molecules-24-02477-t005:** The antioxidative capacity in red apple ciders before and after storage. TE—Trolox equivalent; ABTS— free radical-scavenging ability; FRAP—ferric reducing antioxidative power.

Storage	Type of Yeast	Yeast	Antioxidative Activity (mmol TE/L)	
			ABTS	FRAP
Before storage	*Saccharomyces* *bayanus*	Lalvin C	2.40 ± 0.00^2^	2.60 ± 0.01
		Lalvin QA23	2.30 ± 0.00	2.40 ± 0.00
	*Saccharomyces* *cerevisiae*	Pure Nature	2.20 ± 0.00	2.40 ± 0.00
		Rubino Cru	2.40 ± 0.00	2.50 ± 0.01
		White Arome	2.30 ± 0.00	2.40 ± 0.00
		SIHAFERM Finesse Red	2.50 ± 0.01	2.70 ± 0.01
		SIHAFERM Finesse Red + BCD	2.20 ± 0.00	2.50 ± 0.01
After 3 months of storage	*Saccharomyces* *bayanus*	Lalvin C	2.10 ± 0.00	2.30 ± 0.00
		Lalvin QA23	2.40 ± 0.00	2.30 ± 0.00
	*Saccharomyces* *cerevisiae*	Pure Nature	2.20 ± 0.00	2.40 ± 0.00
		Rubino Cru	2.10 ± 0.00	2.40 ± 0.00
		White Arome	2.40 ± 0.00	2.50 ± 0.01
		SIHAFERM Finesse Red	2.50 ± 0.01	2.60 ± 0.01
		SIHAFERM Finesse Red + BCD	2.40 ± 0.00	2.30 ± 0.00
Type of yeast	*Saccharomyces cerevisiae*		2.32 b ^1^	2.50 a
	*Saccharomyces cerevisiae* + BCD		2.30 c	2.40 b
	*Saccharomyces bayanus*		2.35 a	2.50 a
Storage time	Before storage		2.33 A	2.50 A
	After storage		2.30 A	2.40 B

^1^ a–e: means ± SD followed by different letters within the same line represent significant differences (*p* < 0.05). Data are the averages of triplicates. ^2^ Values are means ± standard deviation; *n* = 3.

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
