# Peer review of "The Influence of Yeast Strain, β-Cyclodextrin, and Storage Time on Concentrations of Phytochemical Components, Sensory Attributes, and Antioxidative Activity of Novel Red Apple Ciders"

_molecules, 2019, doi:10.3390/molecules24132477_

Round 1
Reviewer 1 Report
I believe that this manuscript shows improvements compared to the previous one. But my main concerns remain:
"Overall, the methodology used is not the current and scientifically approved, particularly in the determination of antioxidant activity. The study design is not conveniently described which makes it difficult to interpret and understand the results.
My major concern is about the scope of this journal. I am not sure if this manuscript fits the journal's scope. I am even concerned about the impact and novelty of the work. "
Author Response
Reply for the Reviewer
I believe that this manuscript shows improvements compared to the previous one. But my main concerns remain:
"Overall, the methodology used is not the current and scientifically approved, particularly in the determination of antioxidant activity. The study design is not conveniently described which makes it difficult to interpret and understand the results.
My major concern is about the scope of this journal. I am not sure if this manuscript fits the journal's scope. I am even concerned about the impact and novelty of the work.".
Reply: Thank you very much for your time and your review. The methods has been corrected. The authors has been review, and extend the discussion insisting on the potential beneficial effects of such process, including important role of polyphenols on health. Because this aspect is important for the journal.
Reviewer 2 Report
I have checked the revised manuscript, and I am happy that the points raised by the reviewer have been addressed.
Author Response
Reply for the Reviewer
I have checked the revised manuscript, and I am happy that the points raised by the reviewer have been addressed.
Reply: Thank you very much for your time and your review.
Reviewer 3 Report
The researchers have evaluated the effects of yeast strain (S. bayanus or S. cerevisiae), β-cyclodextrin, and storage time on the development and stability of phytochemicals and sensory attributes in cider from the Bella Marii cultivar of red apple. S. cerevisiae Finesse Red yeast gave ciders with a high content of bioactive compounds and use of β-cyclodextrin during fermentation protected the compounds. Storage time had no significant impact on the tested bioactives (except of the total flavan-3-ols and phenolic acids) or on consumer acceptance. They concluded that that red apple had potential to produce ciders.
The authors have completed a thorough analysis of the composition and stability of red apple ciders. They have provided a valuable source of information. One small weakness of the work is that it is primarily descriptive. The authors could discuss the possible reasons why S. cerevisiae Finesse Red yeast with or without β-cyclodextrin gave the best outcomes.
Ln 155-158 See main comments. This concept should be expanded upon.
Ln 181-182
Ln 196-201 Needs to clarify this statement. One-part talks of no significant differences but then discussion covers differences. Were the differences significant or just tendencies or trends?
Ln 219-223 As for Ln 196-201.
Author Response
Reply for the Reviewer
The researchers have evaluated the effects of yeast strain (S. bayanus or S. cerevisiae), β-cyclodextrin, and storage time on the development and stability of phytochemicals and sensory attributes in cider from the Bella Marii cultivar of red apple. S. cerevisiae Finesse Red yeast gave ciders with a high content of bioactive compounds and use of β-cyclodextrin during fermentation protected the compounds. Storage time had no significant impact on the tested bioactives (except of the total flavan-3-ols and phenolic acids) or on consumer acceptance. They concluded that that red apple had potential to produce ciders.
The authors have completed a thorough analysis of the composition and stability of red apple ciders. They have provided a valuable source of information. One small weakness of the work is that it is primarily descriptive. The authors could discuss the possible reasons why S. cerevisiaeFinesse Red yeast with or without β-cyclodextrin gave the best outcomes.
Reply: Thank you very much for your time and your review. The suggestion has been included in the text.
Ln 155-158 See main comments. This concept should be expanded upon.
Ln 181-182
Reply: These centence has been expanded.
Ln 196-201 Needs to clarify this statement. One-part talks of no significant differences but then discussion covers differences. Were the differences significant or just tendencies or trends?
Ln 219-223 As for Ln 196-201.
Reply: This sentence has been corrected. The reviewer had a right. This sentence has been corrected. The reviewer was right. After re-observing the trials, that the differences significant were in our sample.
This manuscript is a resubmission of an earlier submission. The following is a list of the peer review reports and author responses from that submission.
Round 1
Reviewer 1 Report
The authors described evaluation of phytochemicals components, sensory attributes and antioxidant activity of novel cider from red apple. The experimental results were interesting for the researchers in the related field. However, the manuscript was not well-organized and has many issues to be corrected.
1) Overall, the Table numbering is not accurate. For example, line 58, Table 2 should be Table 3. There are so many same issues in the manuscript. And Table 2 is shown before Table 1. The order of Tables should be corrected.
2) line 263, r2 should be corrected to be superscript.
3) In Table 3, what are the ΔC and h0? I can’t find any clue for the parameters in the manuscript.
4) In table 4, authors reported the SUM of each compound, however we can’t find out the whole SUM of all the compounds. Authors should correct the last compound’s name, “Dihygro”. And I don’t understand what PP and DP meaning are.
5) Figure 1 caption should be correct.
6) Figure 3, F30 -> F3O
7) line 355, 357, cm2, K2S2O5 should be corrected.
8) line 401, (p ≤ 0.05) right? (p < 0.05) ?
9) In conclusion, authors mentioned that Storage time at 4 °C had in universally no effect on 408 the tested quality components of red apple ciders. However, is it right?
The conclusion looks not clear and I don’t understand what authors are trying to conclude.
Reviewer 2 Report
This manuscript pretends to report the impact of different type of yeast, addition β-cycklodextrin and storage time on the physicochemical components and their antioxidant potential of red apple ciders.
Overall, the methodology used is not the must current and scientifically approved, particularly in the determination of antioxidant activity. The study design is not conveniently described which makes it difficult to interpret and understand the results.
It requires an extensive edition of the English language and style.
My major concern is about the scope of this journal. I am not sure if this manuscript fits the journal's scope. I am even concern about the impact and novelty of the work.
Comments:
1. Title
The title, in my opinion, is do not properly reflect the objective of the work.
2. Keywords
The keywords repeats the words of the title. Avoid using the same title words as this will increase the probability that the article will be detected in a search. So, I suggest that authors correct the keywords.
3. Abstract
The abstract is a single paragraph, brief and clear that summarises the content of the article.
I miss a brief introduction.
4. Introduction
The introduction is very brief and not descriptive, lacking information needed to understand the results. I suggest that the authors deepen the subject.
5. Results and Discussion
The results are not organized into an orderly and logical sequence, the fist results presented are in table2?
The discussion contributes to explain the meaning of the results, with support of current and scientifically relevant references.
6. Materials and Methods
The description of the materials and methods is very vague and confusing. The design of the study is unclear, so it is difficult to understand.
- Line 351: “Sample of red apple 'Bella Marii' cultivar were harvested at the Grzegorz Maryniowski 'BioGrim' company in Wojciechów (51°10′22″N 23°03′27″E‐ Poland) at processing maturity during the 2018 year.” This sentence does not make sense.
- Line 357: “To the all must (500 ml) the additive potassium metabisulfite (K2S2O5) and medium in a dose for 0.1 g/L and the yeasts S. cerevisiae (SIHAFERM Pure Nature, SIHA Rubino Cru, SIHA White Arome, SIHAFERM Finesse Red), and S. bayanus (Lalvin C, Lalvin QA 23 YSEO) in a dose of 0.2 g/L were added.” This sentence does not make sense.
- 3.7. Organic acids: The measurements were repeated?
- Lie 391: Add an endpoint to sentence.
- Authors must rewrite item 3.9 and 3.10, both have the same time, it don’t make sense.
7. Conclusions
“What’s more, their SARs were discussed to provide references for the
discovery of anti-inflammatory new drugs.” This sentence is ambiguous and lacking in content.